# HYPERPLR: HYPERGRAPH GENERATION THROUGH PROJECTION, LEARNING, AND RECONSTRUCTION

**Weihuang Wen, Tianshu Yu** *
School of Data Science
The Chinese University of Hong Kong, Shenzhen
`weihuangwen1@link.cuhk.edu.cn, yutianshu@cuhk.edu.cn`

## ABSTRACT

Hypergraphs are essential in modeling higher-order complex networks, excelling in representing group interactions within real-world contexts. This is particularly evident in collaboration networks, where they facilitate the capture of group-wise polyadic patterns, extending beyond traditional pairwise dyadic interactions. The use of hypergraph generators, or generative models, is a crucial method for promoting and validating our understanding of these structures. If such generators accurately replicate observed hypergraph patterns, it reinforces the validity of our interpretations. In this context, we introduce a novel hypergraph generative paradigm, **HyperPLR**, encompassing three phases: **P**rojection, **L**earning, and **R**econstruction. Initially, the hypergraph is projected onto a weighted graph. Subsequently, the model learns this graph's structure within a latent space, while simultaneously computing a distribution between the hyperedge and the projected graph. Finally, leveraging the learned model and distribution, HyperPLR generates new weighted graphs and samples cliques from them. These cliques are then used to reconstruct new hypergraphs by solving a specific clique cover problem. We have evaluated HyperPLR on existing real-world hypergraph datasets, which consistently demonstrate superior performance and validate the effectiveness of our approach.

## 1 INTRODUCTION

Hypergraphs are powerful data structures that go beyond traditional graph theory by enabling a more nuanced representation of multifaceted relationships. Unlike standard graphs, where edges connect merely pairs of nodes, hypergraphs employ hyperedges, each binding multiple nodes, thereby capturing complex, multi-way interactions with higher fidelity. This capability renders hypergraphs indispensable across a vast array of fields such as drug discovery (Kajino, 2019), electronics (Luo et al., 2024), research collaboration networks (Benson et al., 2018), and protein interaction modeling (Feng et al., 2021), where interactions often occur among more than two entities.

Despite the expansive potential of hypergraphs, the generative models that craft these complex structures are still in their nascent stages. Existing models predominantly cater to simpler graph structures or are bound by pre-assumed hypergraph characteristics, which might not be universally applicable to the diverse and intricate patterns present in the actual datasets (Lee et al., 2021; Do et al., 2020; Kook et al., 2020). Recent advancements include learning-based generative models (Zuo et al., 2023; Gailhard et al., 2024) that attempt to sidestep the necessity for predefined data structures; however, these too often fail to encapsulate critical structural nuances that define hypergraph integrity.

In light of these challenges, we propose a pioneering framework known as **HyperPLR** (**P**rojection, **L**earning, and **R**econstruction), which introduces a robust three-phase process for the generation of hypergraphs. Initially, HyperPLR projects the complex hypergraph onto a simplified weighted graph format, essentially maintaining key relationships. It then transitions into a learning phase, wherein it discerns the latent structures within this graph and models the inter-dynamics among the projected entities.

---

*Corresponding author

In the final phase, we define a novel clique cover problem to reconstruct new hypergraphs from the learned weighted projection graph: **Weighted Clique Edge Cover (WCEC)**. It is a common strategy to project hypergraphs onto graphs, but it is evident that this process results in the loss of some higher-order relationships. However, limited research has investigated the reverse direction—reconstructing hypergraphs from graph projections (Bresler et al., 2024; Wang & Kleinberg, 2024). The closest work so far is by Young et al. (2020), which proposes a maximal clique reconstruction that uses the least number of cliques to cover the projected graph, following the principle of parsimony. However, this method does not consider the number of hyperedges or the degree information of vertices. To address these issues, we use a weighted graph as the projection, design the WCEC problem, and show that solving WCEC allows efficient and more controllable hypergraph reconstruction from the projection. Additionally, we develop a fast heuristic for approximating the solution.

Our extensive evaluations of various real-world datasets underscore HyperPLR's adeptness at weaving hypergraphs that faithfully reflect the underlying complexities of group interactions. Thus, HyperPLR marks a significant technological leap in hypergraph generation.

The principal contributions of this paper are encapsulated in the following:

- Introduction and formulation of the WCEC problem, complemented with an efficient algorithmic solution for reconstructing hypergraphs.
- Development of the novel HyperPLR framework, which harmoniously integrates the structure and intricate interactions within a hypergraph.
- Empirical validation of HyperPLR against multiple real-world datasets, showcasing enhanced efficiency and remarkable performance in hypergraph reconstruction.

## 2 RELATED WORK

The study of hypergraph generation has garnered increasing attention as an extension of traditional graph generation techniques, addressing higher-order relational structures present in many real-world applications. Traditional graph generation models have advanced significantly with the introduction of deep learning techniques. (Kipf & Welling, 2016; Jin et al., 2018; Bojchevski et al., 2018) generate graphs by imitating the structural properties of given graphs. This is typically achieved by learning node/edge embeddings and injecting appropriate randomness (Grover et al., 2019; Wang et al., 2019; Dai et al., 2020). However, methods relying on node/edge embeddings often focus primarily on local structures, particularly in large graphs. While graph spectral theory offers an alternative approach, some studies incorporate spectral information into deep models to capture global characteristics (Martinkus et al., 2022; Bojchevski et al., 2018; Rendsburg et al., 2020). Among these, CELL (Rendsburg et al., 2020) stands out for its efficiency and scalability, achieved by removing unnecessary operations from Bojchevski et al. (2018). Another consideration is to generate graphs with weights (edge features) (Grover et al., 2019; Niu et al., 2020; Kocayusufoglu et al., 2022), which again heavily relies on embeddings.

Traditional graph generation models, while effective for molecular graphs and citation networks, are not suitable for hypergraph generation. Hypergraphs, with hyperedges connecting multiple nodes, pose challenges that standard graph models cannot address. Several methods have been proposed for hypergraph generation. HYPERPA (Do et al., 2020) uses a statistical approach, relying on three key metrics from real-world hypergraphs: node count, hyperedge size distribution, and new hyperedges per node. While HYPERPA generates hyperedges based on these metrics, its dependence on precomputed statistics and iterative generation process increases time complexity. HyperDK (Nakajima et al., 2022) generates hypergraphs that preserve local properties of nodes and hyperedges, controlled by two hyperparameters, $d_v$ and $d_e$. A higher $d_v$ value indicates that higher-order information regarding node degrees is preserved, while a higher $d_e$ value implies the preservation of higher-order information regarding hyperedge sizes. ThERA (Kim et al., 2023) organizes nodes in a hierarchical structure across multiple levels, with nodes divided into disjoint levels and deeper levels containing more nodes. It generates hyperedges locally within each group based on a given probability. Compared to parameter-based methods, only a few deep learning models have explicitly addressed hypergraph generation to date (Zuo et al., 2023; Gailhard et al., 2024), and existing learning-based hypergraph generation approaches are limited by scalability issues.

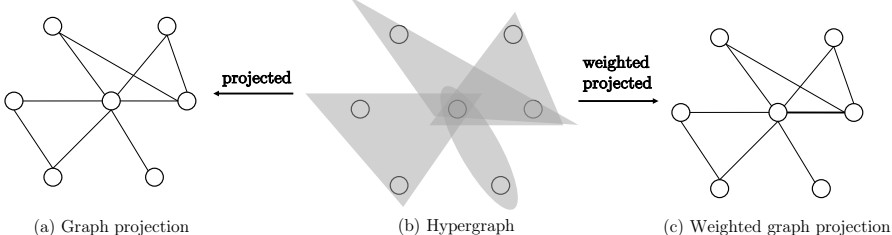

Figure 1: Hypergraph's graph projection (left direction) and weighted graph projection (right direction), with line thickness referring to its weight.

In summary, while deep learning techniques for graph generation have made significant strides, hypergraph generation remains an emerging area of research. Existing models, though promising, often rely on adaptations of graph generation methods and statistical approaches, with inherent limitations in scalability and evaluation. There is a growing need for more robust models specifically designed for hypergraph generation, particularly ones that can efficiently capture the complex, higher-order relationships that hypergraphs represent.

## 3 PRELIMINARIES AND NOTATIONS

**Hypergraph.** A hypergraph is a generalization of a graph where edges, called *hyperedges*, can connect more than two vertices. Formally, a hypergraph is defined as $\mathcal{H} = (V, \mathcal{E})$ where $V$ is a set of vertices and $\mathcal{E}$ is a set of hyperedges $(e_1, e_2, \cdots, e_m)$ with $e_i \subseteq V$. In a hypergraph, hyperedges can be of varying sizes, capturing higher-order relationships between sets of nodes beyond pairwise connections and we assume the hyperedges are distinct in our paper, which means $\forall 1 \leq i, j \leq m, e_i \neq e_j$.

**Graph Projection.** A graph projection, or clique expansion, of a hypergraph, is a transformation process that converts a hypergraph into a traditional graph. In this projection, each hyperedge of the hypergraph is mapped to a clique (a complete subgraph) in the graph, where each pair of nodes in the hyperedge is connected by an edge in the projected graph. Formally, if $\mathcal{H} = (V, \mathcal{E})$ is a hypergraph, the projection graph $G = \text{Proj}(\mathcal{H}) = (V, \mathcal{E}')$, where $\mathcal{E}'$ consists all pairs $(v_i, v_j) \in e_i$ for hyperedge $e_i \in \mathcal{E}$.

**Weighted Graph Projection.** In this paper, we introduce a novel graph projection method for learning-based hypergraph generation, namely Weighted Graph Projection, which effectively preserves the frequency with which vertices co-occur within hyperedges. Formally, given a hypergraph $\mathcal{H} = (V, \mathcal{E})$, the weighted graph projection is to project $\mathcal{H}$ into a weighted graph $G_w = \text{Proj}_w(\mathcal{H}) = (V, \mathcal{E}', w)$. The $w$ is a weight function that assigns a weight to each edge $e' \in \mathcal{E}'$. The common way to assign weight is $w(v_i, v_j) = \sum_{e \in \mathcal{E}} \mathbb{I}(v_i, v_j \in e)$ where $\mathbb{I}$ is an indicator function that equals 1 if both vertices $v_j$ and $v_j$ are part of the same hyperedge.

**Clique Edge Cover Problem.** The Clique Edge Cover problem is concerned with covering all the edges of a graph using the minimum number of cliques (complete subgraphs). This problem is also referred to as the Minimum Clique Edge Cover (MCEC) problem. Given a graph $G = (V, \mathcal{E}')$ a clique cover is a collection of cliques such that every edge in $\mathcal{E}'$ belongs to at least one clique. The goal is to minimize the number of cliques in the cover. This problem is closely related to the projection of hypergraphs, as each hyperedge in a hypergraph corresponds to a clique in the projected graph, and reconstructing the original hyperedges can be seen as a clique cover problem in reverse Bresler et al. (2024).

Fig. 1 illustrates the two different projection methods between hypergraph and traditional graph.

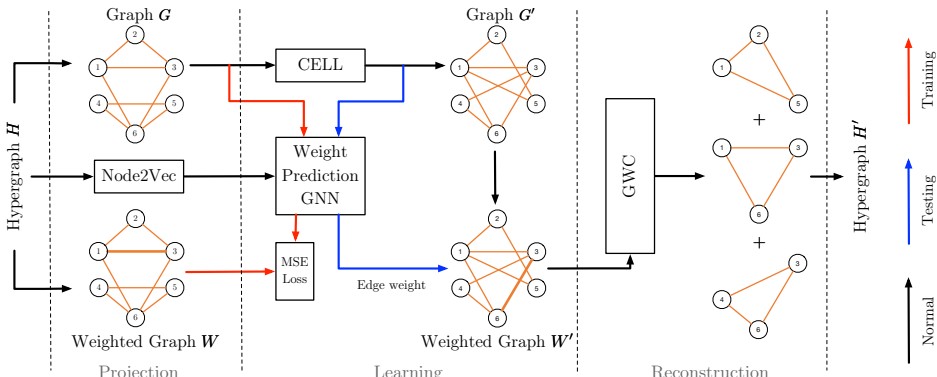

Figure 2: An overview of HyperPLR framework. We use RED and BLUE arrowed lines to indicate flows in the training and the testing phases, respectively. Black arrowed lines are normal flows.

# 4 HYPERPLR

In this part, we elaborate on the framework of HyperPLR in detail. A high-level overview of the HyperPLR framework can be found in Fig. 2. As shown, the framework consists of three core blocks: "**Projection**", "**Learning**", and "**Reconstruction**", acting in the presented order. Given an input hypergraph $\mathcal{H}$, the projection block project $\mathcal{H}$ to the corresponding graph $G$ and weighted graph $G_w$. In the following learning block, we employ CELL (Rendsburg et al., 2020) to sample the new graph $G'$. In parallel to CELL, we adopt a GCN (Kipf & Welling, 2017) to train a weight predictor taking node features of $G$ as input, to optimize the MSE error w.r.t. the real edge weight $w$ in $G_w$. During the testing phase, the sampled graph $G'$ is instead fed into this predictor as the graph topology, and the new weighted graph $G'_w$ is obtained by filling predicted weights into $G'$. In the last block of reconstruction, GWC performs a heuristic strategy to dynamically produce the most likely overlapping cliques out of $G'_w$. Finally, this collection of cliques is readily converted to a new hypergraph $\mathcal{H}'$. We detail each block in the following sections.

## 4.1 PROJECTION

Projecting hypergraphs onto graphs and utilizing graph algorithms is a common technique for solving hypergraph-related problems, as it improves storage efficiency, interpretability, and allows for the use of well-established data structures and algorithms. However, hypergraph projection often results in the loss of higher-order relationships. To address this, there has been research on the limitation when reconstructing hypergraphs from their graph projections (Bresler et al., 2024; Wang & Kleinberg, 2024). Although exact reconstruction of the original hypergraph is generally infeasible, we find that with appropriate "guidance", it is possible to effectively reconstruct most of the patterns in the hypergraph. This insight motivates our investigation into the generation of such "guidance". Specifically, we demonstrate that learning to generate this guidance can enable the creation of a new hypergraph that retains most of the information from the original hypergraph's projection. In our study, we show that the weighted graph projection $G_w$ serves as an effective form of guidance during the reconstruction process.

On one hand, the weighted projection offers a more condensed hyperedge information into edges, reducing graph size compared to bipartite representations, which require separate hyperedge nodes. As discussed in Bojchevski et al. (2018); Rendsburg et al. (2020), bipartite graphs are difficult to analyze using spectral tools, as they do not possess steady states. On the other hand, the weighted projection provides a higher representational capacity than the unweighted projection. Notably, in the weighted projection, the weight of each edge in the graph reflects the co-occurrence frequency of its nodes within hyperedges, thereby preserving crucial information for the reconstruction of the original hypergraph.

## 4.2 LEARNING

Learning to generate weighted projection $G_w$ that resembles the projection from the real-world hypergraph is non-trivial, as 1) $G_w$ with weights are hard to model and learn, and 2) real-world hypergraph can easily scale up (to tens of thousands of vertices) with long-range dependencies like overlapping cliques or communities in large hypergraphs. We first present how this can be achieved at a high level. Considering a weighted graph with node feature $X \in \mathbb{R}^{n \times d}$, adjacency matrix $A$, and weight matrix $W$, we have:

$$\mathbb{P}(W, A|X) = \mathbb{P}(W|A, X)\mathbb{P}(A) \tag{1a}$$

$$= \mathbb{P}(W|A, X) \prod_{i,j} \mathbb{P}(A_{i,j}) \tag{1b}$$

$$= \prod_{i,j} \mathbb{P}(W_{i,j}|A_{i,j}, X) \prod_{i,j} \mathbb{P}(A_{i,j}) \tag{1c}$$

$$= \prod_{i,j} \mathbb{P}_\alpha(W_{i,j}|A_{i,j}, X_i, X_j)\mathbb{P}_\beta(A_{i,j}), \tag{1d}$$

where $\mathbb{P}(\cdot)$ measures the probability. Eq. (1a) stands because $A$ is derived solely from a spectral method, independent of $X$. Eq. (1b) is due to the fact that $A$ is drawn from an edge-independent "score matrix" as in Bojchevski et al. (2018); Rendsburg et al. (2020). Eq. (1c) and (1d) are further built upon the assumption that edge weight $W_{ij}$ is only dependent on the features of the ending nodes $X_i$ and $X_j$, and the existence of this edge $A_{i,j}$, under two distributions parameterized by $\alpha$ and $\beta$.

We first introduce $\mathbb{P}_\beta(\cdot)$ allowing for generating large-scale graphs with high efficiency. To this end, we employ CELL (Rendsburg et al., 2020). CELL is a graph generative model working in the spectral space by removing redundant computation from NetGAN (Bojchevski et al., 2018). In NetGAN, the following objective is optimized

$$\min_{Z \in \mathbb{R}^{n \times n}} - \sum_{(u,v) \in \mathcal{R}} \log \sigma_{\text{rows}}(Z)_{u,v} \qquad \text{s.t.} \quad \text{rank}(Z) \leq h, \tag{2}$$

where $\mathcal{R}$ is a collection of (massive) random walks and $(u, v)$ is a transition. $\sigma_{\text{rows}}(Z)$ is a row-wise softmax function perform on the low-rank variable $Z$. In practice, NetGAN needs to sample millions of random walks from a graph with moderate size, which is prohibitively expensive. Besides, the extra computational burden is required as NetGAN trained a generator to produce fixed-length random walks to obtain a new graph.

CELL (Rendsburg et al., 2020) argued that such a massive amount of sampling procedures and the generator in NetGAN can be avoided by only considering the spectral properties in limit. Concretely, given the adjacency matrix $A$ of a graph, its transition matrix can be obtained via $P = D^{-1}A$ where $D = \text{diag}(d) \in \mathbb{R}_+^{n \times n}$ is the degree matrix. Assuming the stationary state $\pi$ of $P$ exists and is unique, CELL considers acting an infinite number of infinite long random walks and calculates how many times a node can be visited. The time of visit for nodes can be encoded in a "score matrix" $S$, and the normalized score matrix in limit is

$$\lim_{\substack{q \to \infty \\ T \to \infty}} \frac{S}{qT} = \text{diag}(\pi)P, \tag{3}$$

where $q$ and $T$ are number and length of random walks, respectively. Thus $S$ can serve as a surrogate of random walk set $\mathcal{R}$ in Eq. (2):

$$\min_{Z \in \mathbb{R}^{n \times n}} - \sum_{u,v=1}^{n} S_{u,v} \log \sigma_{\text{rows}}(Z)_{u,v}. \tag{4}$$

Since $\text{diag}(\pi)P \propto A$ and rescaling $S$ will not change the optima in Eq. (4), the final objective of CELL becomes

$$\min_{Z \in \mathbb{R}^{n \times n}} - \sum_{u,v=1}^{n} A_{u,v} \log \sigma_{\text{rows}}(Z)_{u,v} \qquad \text{s.t.} \quad \text{rank}(Z) \leq h, \tag{5}$$

whose solution is denoted as $Z^*$. Then $P^* = \sigma_{\text{rows}}(Z^*)$ can be viewed as a low-rank-regularized transition matrix. We can obtain a new score $S^* = \text{diag}(\pi^*)P^*$, where $\pi^*$ is the stationary distribution of $P^*$. Finally, the edges of a new graph can be independently sampled according to $S^*$.

Next, we discuss $\mathbb{P}_\alpha(\cdot)$. To allow generating graphs with weights, we extend CELL by providing predicted weights on the sampled edges, where the edge weight predictor is a Graph Convolutional Network (GCN) (Kipf & Welling, 2017). During the training phase, this predictor takes the vertex embeddings $X^{(0)}$ and the adjacency matrix $A$ as input, and outputs a set of new embedding $X^{(l)}$:

$$X^{(l)} = \text{GCN}(X^{(0)}, A), \tag{6}$$

where $l$ is the number of basic GCN layers. The loss is the MSE between the predicted weights $X^{(l)}X^{(l)\top}$ and the ground-truth weights $W$ of the original WLIG on the observed edge set $A$:

$$\sum_{(i,j)\in A} \left( \left( X^{(l)}X^{(l)\top} \right)_{i,j} - W_{i,j} \right)^2. \tag{7}$$

This weight predictor can be trained not only on a single graph projection (though we train separate embedding networks for each instance) but throughout a set of formulas, since all these formulas may share common structures on weights. In the testing stage, we feed the embedding $X^0$ and a sampled adjacency $A'$ from CELL into the well-trained GCN:

$$X_s^{(l)} = \text{GCN}(X^{(0)}, A'). \tag{8}$$

Then weights can be derived using $X_s^{(l)}X_s^{(l)\top}$. We only consider the weights on the edges that appear in $A'$. See the "Learning" block in Fig. 2 for an intuitive view.

One remaining problem is how to obtain the initial Node Embedding $X^{(0)}$ for each vertex. One of the most intuitive ways is to use a one-hot representation, but as the size of the hypergraph grows, the vertices number $n$ and the required dimension increase rapidly. In practice, we train the node embedding with the Node2Vec algorithm (Grover & Leskovec, 2016) from the simple projection graph $G$.

## 4.3 RECONSTRUCTION

As each hyperedge in the hypergraph $\mathcal{H}$ is mapped to a clique in the graph projection $G$ or $G_w$, it is straightforward to consider the reconstruction as a clique cover problem, for example, MCEC problem (Rodrigues, 2021). Here, we give a proposition about the limitations of the MCEC:

**Proposition 1.** *One hypergraph $\mathcal{H}$ can be reconstructed by solving the MCEC on the graph projection $G$ if and only if each hyperedge appears as an isolated clique.*

*Proof.* A hyperedge must be a clique in the graph projection, but not vice versa. A clique may not be a hyperedge in only two cases: 1) for a hyperedge $e$, there is a hyperedge $e'$ set that $e \subset e'$; 2) for several hyperedges $e_1, \cdots, e_k$, which's projections overlap together and form a bigger clique than each other. The first case would lead the solving of MCEC to ignore some hyperedges which are included by other hyperedges. The second case would lead the MCEC solving to produce the new hyperedges. □

Although the MCEC has its limitations, it still can be a strong baseline in the hypergraph reconstruction task (Young et al., 2020). Due to the "principle of parsimony", solving MCEC on the projection graph will ignore the frequency of node level pairwise appearance and always recover fewer hyperedges than the original hypergraph. To address these issues, inspired by MCEC, we make a step to raise a novel Weighed Clique Edge Cover (WCEC) problem and an associated efficient algorithm Greedy Weighted Cover (GWC) in a hill-climbing manner. WCEC substantially differs from MCEC by taking into account an essential but less explored factor–frequency of node level pairwise appearance.

**Weighted Clique Edge Cover Problem (WCEC).** Though there exist several variants of clique covering problems, our problem of recovering clique covers from weighted graphs has been less studied. The most related problem in literature is *Weighted Edge Clique Partition (WECP)*, which is

proposed by Feldmann et al. (2020) and we provide a more comprehensive analysis for the connection and difference betwen WECP and WCEC in Appendix. A. Here, we define the WCEC problem as follows:

**Problem 1** (Weighted Clique Edge Cover)**.** *Given a weighted graph $\mathcal{G}$ with weight matrix $W$, it asks to select $k$ cliques from $\mathcal{G}$, such that the distance $d(W, W') \geq 0$ between $W$ and $W'$ is miminized, where weight matrix $W'$ is derived from a new weighted graph $\mathcal{G}'$ by stacking $k$ selected cliques.*

In our setting, we let $d(W, W')$ be the L1-distance:

$$d(W, W') = \sum_{i=1}^{n} \sum_{j=1}^{n} |W_{ij} - W'_{ij}|, \tag{9}$$

where $W_{ij}$ represents the entry of $W$ at $i$th row and $j$th column. Comparing the proposed WCEC and MCEC, we have the following lemma:

**Lemma 2.** *Suppose a WCEC problem with weight matrix $W$ is obtained by assigning positive weights to edges in adjacency $A$ associated with an MCEC problem. Then a solution clique cover to this WCEC such that $d(W, W') = 0$ must form a cover of the MCEC, but a clique cover of MCEC on $A$ may not satisfy $d(W, W') = 0$.*

*Proof.* We first prove the initial part. By definition, $d(W, W') = 0$ implies that for every $(i, j)$, the weight matrix $G_w$ of the found cover satisfies $W'_{i,j} = W_{i,j}$. Since $W$ is derived by assigning positive weights to the non-zero entries in $A$, we have $W'_{i,j} > 0 \implies A_{i,j} = 1$ and $W_{i,j} = 0 \implies A_{i,j} = 0$. Therefore, the solution to the WCEC is a cover of the MCEC. For the second part, it suffices to provide an easily verifiable example, which is presented as a constructive proof in Appendix. B. □

Lemma 2 provides an interpretation of why the proposed framework "weighted projection +WCEC" is more powerful than "projection + MCEC" utilized by Young et al. (2020). In general, solving WCEC provides rich information for solving MCEC, but not vice versa. Nevertheless, solving either of them is NP-complete (Feldmann et al., 2020; Ullah, 2022).

**Greedy Weight Cover (GWC) Algorithm.** Due to NP-completeness, an exact solution to WCEC can be infeasible. Therefore, we develop a specialized algorithm to obtain a local optima of WCEC via hill-climbing. Our algorithm works on a local search basis, wherein each iteration will take action with the largest gain. We term this procedure the Greedy Weighted Cover (GWC) algorithm. Concretely, GWC first enumerates or samples the cliques set from the weighted graph projection, whose sizes are below the maximal hyperedge size of the original hypergraph, using the algorithm proposed by Zhang et al. (2005). Then we create a `GainTable` measuring how much one can improve by decoding the corresponding clique. In each subsequent step, GWC finds a clique with the largest gain and updates the `GainTable` in a dynamic programming fashion, until the number of found cliques reaches a preset threshold $m$ or distance. To prevent redundant computation, we first build an `EdgeTable` mapping from an edge to its related cliques (see Alg. 2), which allows GWC only updates the gain of the *local cliques* in each step. The overall procedure is summarized in Alg. 1.

## 5 EXPERIMENT

### 5.1 PROTOCALS

**Implementation details.** Throughout all the experiments, HyperPLR is running on an Apple M1 CPU. The dimension of node embedding from Node2Vec is 50. The GCN consists of two layers, with input/output dimensions 50/128 and 128/128, respectively. We employ ADAM as the optimizer for all learning modules (i.e., Node2Vec, GCN, and CELL). For CELL, we set parameter `edge_overlap_limit` = 0.8 which controls the overlapping portion of the generated and the original graphs. For more details, one can refer to the supplementary material.

---

**Algorithm 1:** Greedy Weight Coverage

---

1    **Input:** weighted adjacency matrix $W$, set of cliques $\mathcal{C}$, clause number $m$
     **Output:** set of chosen cliques `Cover`
2    **function** GWC($W, \mathcal{C}, m$)
3      `Cover` $\leftarrow \varnothing$
4      `GT` $\leftarrow \{C_1 : g_1, \cdots, C_n : g_n\}$ `// Gain table`
5      $\mathcal{E} \leftarrow$ GenerateEdgeTable($\mathcal{C}$)
6      **for** $k \leftarrow 1$ *to* $m$ **do**
7        $C' \leftarrow$ LargestGain(`GT`)
8        `Cover` $\leftarrow$ `Cover` $\cup\, C'$
9        `GT`$[C'] \leftarrow -\infty$
10       $W \leftarrow$ Update($W, C'$)
11       **for** *Edge* $e \in C'$ **do**
12         `Cliques` $\leftarrow \mathcal{E}[e]$
13         **for** $C \in$ `Cliques` **do**
14          `GT`$[C] \leftarrow$ Update$'(W, C)$
15      **return** `Cover`

---

**Algorithm 2:** Generate Edge Table

---

1    **Input:** set of cliques $\mathcal{C}$
     **Output:** edge table $\mathcal{E}$
2    **function** GenerateEdgeTable($\mathcal{C}$)
3      $\mathcal{E} \leftarrow \{e_1 : \varnothing, \cdots, e_n : \varnothing\}$ `// Edge table`
4      **forall** $e \in \mathcal{E}, C \in \mathcal{C}$ **do**
5        **if** $e \in C$ **then**
6         $\mathcal{E}[e] \leftarrow \mathcal{E}[e] \cup C$
7      **return** $\mathcal{E}$

---

**Dataset.** Following the standard approach outlined in Nakajima et al. (2022); Lee et al. (2021); Kim et al. (2023), we evaluate performance on five datasets across three domains: contact (contact-high-school, contact-primary-school), email (email-Eu, email-Enron), and drug (NDC-classes) (Benson et al., 2018). In the contact datasets, each node represents an individual, and each hyperedge represents a group communication event involving all participating individuals. In the email datasets, each node represents a user, and each hyperedge corresponds to an email, consisting of the sender and all recipients. In the drug dataset, each hyperedge corresponds to an NDC code for a drug, and the nodes represent the substances composing the drug. More details can be found in Appendix. C.1.

**Baselines.** We compared the performance of our proposed model against four existing hypergraph generation methods: HyperDK (Nakajima et al., 2022), Hyperlap (Lee et al., 2021), Hyperlap+ (Lee et al., 2021), and THERA (Kim et al., 2023). HyperDK, Hyperlap, and Hyperlap+ are static generators, primarily suited for producing static hypergraphs, while THERA generates dynamic graphs, such as temporal hypergraphs. Since HyperDK is highly sensitive to parameter settings, we evaluated it under two configurations: $\text{HyperDK}_{0,0}$, which aims to generate hypergraphs that preserve the average node degree and the average hyperedge size, and $\text{HyperDK}_{1,1}$, which seeks to preserve the degree of each individual node and the size of each hyperedge.

## 5.2 RESULTS

**Graph statistics.** The results of the comparisons between our model and state-of-the-art methods across five datasets, using various evaluation metrics, are presented in Table 1. We further compare some of the advanced structural properties in Appendix. C.2. Each dataset was generated five times, and the average results were reported. The graph properties evaluated include hypergraph structure-level metrics (density, average hyperedge size, average node degree), graph projection-level metrics ("G coefficient" and "G modularity"), and bipartite graph representation-level metrics ("B modularity"). These properties provide a comprehensive overview of the graph's structural characteristics. The detailed definitions about the metrics can be found in Appendix. C.3.

**Representation ability and reconstruction ability.** We investigate how much the graph structure statistics can retain after feeding a hypergraph into different consecutive phases of HyperPLR from projection $G_w$, learning $G_w^L$, and reconstruction $G_w^R$. Results are in Table 2. We also visualize this process using the `email-Enron` instance in Fig. 3. It is evident that the main structures of the WLIGs are well-preserved across all three phases.

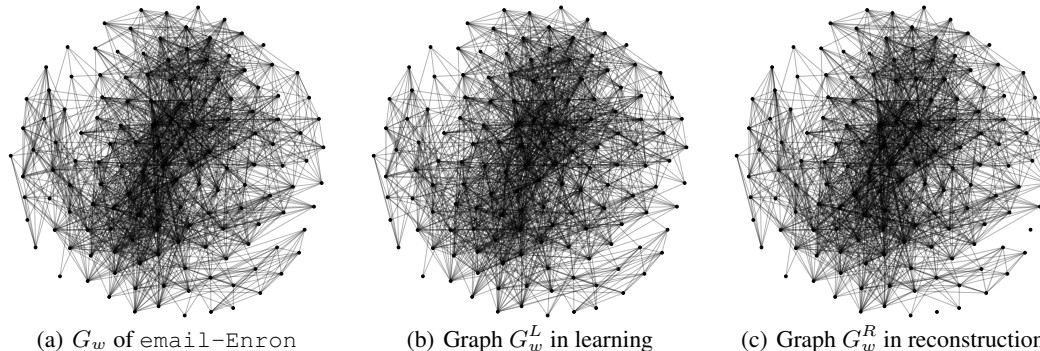

(a) $G_w$ of email-Enron     (b) Graph $G_w^L$ in learning     (c) Graph $G_w^R$ in reconstruction

Figure 3: Visualization of weighted graph projection of hypergraph email-Enron at different phases. Each node corresponds to one vertex and each edge (with thickness) indicates inter-node incidence (with frequency). (a) Weighted graph projection $G_w$ derived from original hypergraph; (b) Generated Weighted graph projection $G_w^L$ based on (a) using hyperPLR; (c) Weighted graph projection $G_w^R$ of generated hypergraph from (b) with GWC. Zoom in for a better view.

Table 1: Results of hypergraph generation in terms of graph metrics with 5 real-world hypergraphs

| | density | average size | average degree | G coefficient | G modularity | B modularity |
|---|---|---|---|---|---|---|
| **contact-high-school** | 23.908 | 2.327 | 55.633 | 0.504 | 0.582 | 0.754 |
| $\text{HyperDK}_{0,0}$ | 272.161 | 3.089 | 840.796 | 0.999 | 0.000 | 0.324 |
| $\text{HyperDK}_{1,1}$ | 142.573 | 2.178 | 310.567 | 0.861 | 0.016 | 0.459 |
| Hyperlap | 23.938 | **2.327** | 55.701 | 0.341 | 0.394 | 0.622 |
| Hyperlap+ | **23.908** | **2.327** | **55.633** | 0.632 | 0.742 | **0.747** |
| THERA | **23.908** | 2.609 | 62.382 | 0.354 | 0.405 | 0.831 |
| **HyperPLR** | 24.055 | 2.703 | 65.011 | **0.471** | **0.556** | 0.649 |
| **contact-primary-school** | 52.496 | 2.419 | 126.979 | 0.526 | 0.284 | 0.635 |
| $\text{HyperDK}_{0,0}$ | 229.896 | 3.129 | 719.429 | 1.000 | 0.000 | 0.319 |
| $\text{HyperDK}_{1,1}$ | 143.343 | 2.285 | 327.516 | 0.924 | 0.010 | 0.437 |
| Hyperlap | **52.496** | **2.419** | **126.979** | **0.496** | **0.281** | 0.620 |
| Hyperlap+ | **52.496** | **2.419** | **126.979** | 0.655 | 0.496 | **0.640** |
| THERA | **52.496** | 2.767 | 145.252 | 0.446 | 0.150 | 0.758 |
| **HyperPLR** | 52.714 | 2.648 | 139.593 | 0.491 | **0.281** | 0.600 |
| **email-Enron** | 10.573 | 3.009 | 31.818 | 0.593 | 0.352 | 0.656 |
| $\text{HyperDK}_{0,0}$ | 52.045 | 3.188 | 165.933 | 0.958 | 0.007 | 0.322 |
| $\text{HyperDK}_{1,1}$ | 39.596 | 2.885 | 114.252 | 0.796 | 0.040 | 0.355 |
| Hyperlap | 10.232 | **3.131** | 32.034 | 0.646 | 0.275 | 0.576 |
| Hyperlap+ | 10.203 | **3.131** | **31.944** | 0.841 | 0.582 | 0.738 |
| THERA | 10.203 | 3.386 | 34.545 | 0.674 | 0.166 | **0.736** |
| **HyperPLR** | **10.723** | 3.244 | 34.782 | **0.593** | **0.351** | 0.569 |
| **email-Eu** | 25.077 | 3.426 | 85.909 | 0.492 | 0.371 | 0.675 |
| $\text{HyperDK}_{0,0}$ | 156.130 | 3.125 | 487.965 | 0.723 | 0.015 | 0.327 |
| $\text{HyperDK}_{1,1}$ | 114.040 | 2.887 | 329.223 | 0.788 | 0.035 | 0.353 |
| Hyperlap | **25.778** | **3.621** | 93.338 | 0.623 | 0.280 | **0.571** |
| Hyperlap+ | 25.860 | **3.621** | 93.633 | 0.785 | 0.745 | 0.751 |
| THERA | 24.868 | 4.696 | 116.792 | 0.837 | 0.099 | 0.793 |
| **HyperPLR** | 29.081 | 3.135 | **91.155** | **0.446** | **0.319** | 0.493 |
| **NDC-classes** | 0.937 | 5.922 | 5.550 | 0.611 | 0.610 | 0.741 |
| $\text{HyperDK}_{0,0}$ | 36.112 | 3.583 | 129.392 | 0.311 | 0.038 | 0.303 |
| $\text{HyperDK}_{1,1}$ | 31.083 | 3.594 | 111.711 | 0.754 | 0.058 | 0.296 |
| Hyperlap | 1.089 | **6.166** | 6.718 | **0.672** | 0.456 | 0.575 |
| Hyperlap+ | 1.098 | **6.166** | 6.767 | 0.750 | 0.793 | 0.836 |
| THERA | **0.999** | 6.282 | **6.277** | 0.793 | 0.520 | 0.892 |
| **HyperPLR** | 1.210 | 5.501 | 6.655 | 0.791 | **0.623** | **0.733** |

## 5.3 ANALYSIS AND DISCUSSION

The experimental results demonstrate several key findings regarding the performance and capabilities of the proposed HyperPLR framework:

**Graph Statistics Comparison.** Towards graph-level metrics, HyperPLR outperformed traditional hypergraph generation models, including HyperDK, Hyperlap, and ThERA, across various datasets. Remarkably, despite not receiving any direct structural parameters from the original hypergraph,

Table 2: Results of the graph metrics for the weighted graph projection $G_w$ derived from original hypergraph; Generated Weighted graph projection $G_w^L$; and weighted graph projection $G_w^R$ of generated hypergraph

| Hypergraph | $G_w$ coefficient | $G_w$ modularity | $G_w^L$ coefficient | $G_w^L$ modularity | $G_w^R$ coefficient | $G_w^R$ modularity |
|---|---|---|---|---|---|---|
| contact-high-school | 0.504 | 0.582 | 0.388 | 0.546 | 0.472 | 0.556 |
| contact-primary-school | 0.526 | 0.284 | 0.453 | 0.308 | 0.491 | 0.277 |
| email-Enron | 0.593 | 0.352 | 0.508 | 0.433 | 0.593 | 0.347 |
| email-Eu | 0.492 | 0.371 | 0.366 | 0.448 | 0.446 | 0.315 |
| NDC-classes | 0.611 | 0.610 | 0.452 | 0.538 | 0.788 | 0.625 |

HyperPLR consistently demonstrates competitive similarity across nearly all graph metrics when compared to other parameter-based methods, underscoring its superior potential in hypergraph generation. Additionally, the clustering coefficient distribution of the hypergraphs generated by HyperPLR exhibits the highest degree of similarity to real-world datasets. Furthermore, HyperPLR is parameter-insensitive, delivering stable and robust results with default parameter settings. For example, the default rank number of CELL Rendsburg et al. (2020) is 9. To investigate how much this parameter impacts the generated results, we conduct extra experiments on varying $\mathrm{rank} = 8, 10, 12$. We also present three generated weighted graph projections in Appendix. D. Additionally, a brief discussion on the trade-off between the fidelity and diversity of HyperPLR is provided in Appendix. E. All these reuslt indicate that the HyperPLR framework, despite not requiring parameter tuning specific to each dataset, can generalize effectively across different domains. This robustness and parameter insensitivity highlight HyperPLR's advantage in practical applications, where real-world datasets can vary widely in structure.

**Representation and Reconstruction Ability.** Table 2 and Figure 3 provide insights into how HyperPLR maintains structural properties throughout the projection, learning, and reconstruction phases. The close similarity between the weighted graph projections derived from the original hypergraph ($G_w$) and the generated hypergraph ($G_w^R$) demonstrates that HyperPLR effectively preserves critical structural information. The consistency in clustering coefficients and modularity across the different phases (as seen in Table 2) suggests that the proposed weighted graph projection method captures essential high-order relationships. This is further confirmed by the visualization in Figure 3, where the main structures are visibly well-retained, implying that the framework's learning and reconstruction processes are efficient in preserving hypergraph topology.

**Limitations and Future Directions.** Despite its strengths, HyperPLR has limitations. It requires significant computational resources, especially for larger datasets, and the GWC heuristic algorithm, while effective, doesn't guarantee optimality. Future research could explore advanced optimization techniques like reinforcement learning or MCTS for improving clique selection. Additionally, the complexity of HyperPLR is constrained by the exponential nature of the maximal clique enumeration problem, though it remains manageable for real-world hypergraphs. Future work could also focus on accelerating clique enumeration using hyperedges as prior information.

## 6 CONCLUSION

Deep hypergraph generation models have demonstrated their ability to capture the complex distribution of hypergraphs by leveraging the rich information embedded within them to generate more realistic structures. In this paper, we address a novel clique cover problem—WCEC—for reconstructing hypergraphs from weighted graph projections, and we propose an efficient and effective framework, HyperPLR, for generating real-world hypergraphs. The weighted graph projection shows strong representational power compared to existing inexact graph representations, while also significantly enhancing learning and generation against the exact representation. This is the first work to achieve both. Our method demonstrated superior efficiency and stability compared to previous state-of-the-art methods across various graph-based metrics. In the future, we aim to extend HyperPLR's capabilities to additional generation tasks, such as bipartite network generation or SAT instance generation, in order to capture the rich information inherent in these structures.

ACKNOWLEDGMENTS

This work was supported by the National Science and Technology Major Project of China under Grant 2022ZD0116408. This work is also supported by the Guangdong Provincial Key Laboratory of Mathematical Foundations for Artificial Intelligence (2023B1212010001). We would like to thank all the anonymous reviewers for their insightful comments. We also sincerely thank Chaolong Ying and Huizhi Zhu for their valuable discussions, which greatly contributed to the development of this work.

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

## A  WECP PROBLEM

**Problem 2** (Weighted Edge Clique Partition). *Given a graph $\mathcal{G}$ with weight function $w_e : E(G) \to \mathbb{Z}^+$ and a positive integer $k$, it asks to select at most $k$ cliques such that each edge appears in exactly as many cliques as its weight (if it exists, otherwise output **NO**).*

Note that although the WECP may appear similar to the WCEC problem, they have essential differences. In WECP, the objective is to partition the edges such that each edge is covered exactly as many times as indicated by its given weight. This means that if an edge has a weight greater than one, it must be covered by multiple cliques, and no edge can be covered more or fewer times than specified. This strict covering constraint makes the problem more restrictive. The solution either finds such a partition or determines that it is impossible, framing WECP as a Constraint Satisfaction Problem (CSP).

In contrast, the WCEC problem aims to minimize the distance between the original weight matrix $W$ of the graph and a derived weight matrix $W'$, which is formed by stacking the selected cliques. In this sense, it approximates or covers the edge weights using cliques. Unlike WECP, there is no strict requirement for how many times each edge must be covered; instead, the focus is on minimizing the difference between the original and derived weight distributions. Therefore, WCEC is essentially a combinatorial optimization problem.

## B  ADDITIONAL PROOF FOR THE SECOND PART OF LEMMA 2

*Proof.* Consider a simple hypergraph $\mathcal{H} = (V, \mathcal{E})$ where $V = \{v_1, v_2, v_3, v_4\}$ and $\mathcal{E} = \{(v_1, v_2, v_4), (v_2, v_3, v_4), (v_1, v_3)\}$. The graph projection $G$ of $\mathcal{H}$ will be a complete graph. If $G$ is used as the input for the minimum clique edge cover problem, the optimal solution will be a clique $c = (v_1, v_2, v_3, v_4)$. In the WCEC setting, let $W$ be the weighted matrix of the weighted graph projection of hypergraph $\mathcal{H}$, and let $W'$ be the weighted matrix derived from the solution of the MCEC. Then we have:

$$W = \begin{bmatrix} 0 & 1 & 1 & 1 \\ 1 & 0 & 1 & 2 \\ 1 & 1 & 0 & 1 \\ 1 & 2 & 1 & 0 \end{bmatrix}, \qquad W' = \begin{bmatrix} 0 & 1 & 1 & 1 \\ 1 & 0 & 1 & 1 \\ 1 & 1 & 0 & 1 \\ 1 & 1 & 1 & 0 \end{bmatrix}.$$

It is evident that $d(W, W') = \sum_{i=1}^{n} \sum_{j=1}^{n} |W_{ij} - W'_{ij}| = 2$, which is not the optimal solution for WCEC problem. $\qquad \square$

## C  EXPERIMENTAL DETAILS

The code of our experiments is publicly available at `https://github.com/LOGO-CUHKSZ/HyperPLR`.

### C.1  DATASETS

Our experimental evaluation utilizes five real-world datasets from Benson et al. (2018):

- **contact-high-school**: This dataset is constructed from interactions recorded by wearable sensors at a high school, consisting of 327 nodes, 172,035 timestamped hyperedges, and 7,818 unique hyperedges. The projected graph contains 5,818 edges and 6,151 maximum cliques.
- **contact-primary-school**: This dataset is constructed from interactions recorded by wearable sensors at a primary school, consisting of 242 nodes, 106,879 timestamped hyperedges, and 12,704 unique hyperedges. The projected graph contains 8,317 edges and 100,153 maximum cliques.
- **email-Enron**: In this dataset, nodes represent email addresses at Enron, and each hyperedge comprises the sender and all recipients of an email. The dataset contains 143 nodes,

Table 3: Comparative result of advanced properties of the ground-truth hypergraphs and generation hypergraphs.

| Diameter | contact-high-school | contact-primary-school | email-Enron | email-Eu | NDC-classes |
|---|---|---|---|---|---|
| $\text{HyperDK}_{0,0}$ | 0.500 | 0.333 | 0.500 | 0.667 | 0.778 |
| $\text{HyperDK}_{1,1}$ | 0.250 | 0.333 | 0.250 | 0.333 | 0.533 |
| Hyperlap | **0.200** | 0.067 | 0.250 | 0.333 | 0.422 |
| Hyperlap+ | 0.250 | 0.333 | 0.050 | 0.133 | 0.289 |
| TheRA | 0.250 | **0.000** | 0.250 | 0.667 | 0.222 |
| HyperPLR | 0.250 | **0.000** | **0.000** | **0.033** | **0.000** |
| **Triangles Num** | | | | | |
| $\text{HyperDK}_{0,0}$ | 167.117 | 21.456 | 41.427 | 156.738 | 184.785 |
| $\text{HyperDK}_{1,1}$ | 84.440 | 15.988 | 10.867 | 41.227 | 79.077 |
| Hyperlap | 0.599 | 0.409 | 3.315 | 8.998 | 3.536 |
| Hyperlap+ | 0.942 | 0.284 | 1.517 | 2.981 | 2.124 |
| TheRA | 0.468 | **0.198** | 2.451 | 3.725 | 1.638 |
| HyperPLR | **0.316** | 0.354 | **0.055** | 0.742 | **0.281** |
| **Degree Distribution** | | | | | |
| $\text{HyperDK}_{0,0}$ | 23.026 | 23.026 | 23.026 | 23.026 | 23.026 |
| $\text{HyperDK}_{1,1}$ | 21.002 | 22.635 | 12.406 | 8.490 | **3.960** |
| Hyperlap | 5.958 | 8.072 | 9.294 | 6.946 | 4.415 |
| Hyperlap+ | 5.336 | 8.812 | 9.023 | **6.448** | 4.474 |
| TheRA | 6.634 | 10.530 | 7.986 | 15.315 | 14.325 |
| HyperPLR | **5.065** | **7.657** | **4.639** | 7.424 | 4.020 |
| **Singular Value Distribution** | | | | | |
| $\text{HyperDK}_{0,0}$ | 6.212 | 5.730 | 4.653 | 6.422 | 6.750 |
| $\text{HyperDK}_{1,1}$ | 5.462 | 5.147 | 4.513 | 6.345 | 7.008 |
| Hyperlap | 5.369 | 5.043 | 4.470 | 6.293 | 6.589 |
| Hyperlap+ | 5.372 | 5.056 | 4.520 | 6.319 | 6.560 |
| TheRA | 5.377 | 5.054 | 4.465 | 6.339 | 6.620 |
| HyperPLR | **5.347** | **5.032** | **4.451** | **6.278** | **6.432** |

10,883 timestamped hyperedges, and 1,512 unique hyperedges. The projected graph contains 1,800 edges and 10,883 maximum cliques.

- **email-Eu**: This dataset includes email addresses at a European research institution, with hyperedges representing the sender and all recipients of an email with the same timestamp. The dataset consists of 998 nodes, 234,760 timestamped hyperedges, and 25,027 unique hyperedges. The projected graph contains 29,299 edges and 237,231 maximum cliques.

- **NDC-classes**: In this dataset, each hyperedge corresponds to a drug, and the nodes are the class labels assigned to the drugs. The dataset consists of 1,161 nodes, 49,724 timestamped hyperedges, and 1,088 unique hyperedges. The projected graph contains 6,222 edges and 624 maximum cliques.

## C.2 ADVANCED PROPERTIES

Inspired by the reviewer, we expand the evaluation metrics to some advanced properties like **SHyRe** in Wang & Kleinberg (2024). The result is presented in Table 3. Notably, the simplicial closure metric from **SHyRe** evaluates the temporal evolution process of dynamic hypergraphs, is not applicable to our work or the baseline static generation models. Therefore, we use a similar concept, triangle numbers, which is a widely used graph structural metric (Sansford et al., 2023). For density, we have already presented the results in Table 1, so it is not repeated here. Similar to the approach in **SHyRe**, we standardized the metrics for triangle numbers and diameter using the formula: $\frac{|x_1 - x_2|}{x_1}$, where $x_1$ and $x_2$ represent the values of the ground truth and the generated graphs, respectively. For degree distribution and singular value distribution, we compared their cross-entropy. A smaller value indicates better alignment for all the metrics. From the results, we observe that HyperPLR outperforms or is competitive with other models across most datasets and metrics.

## C.3 EVALUATION METRICS

We evaluate the training and generated hypergraphs using the following six metrics:

- **Density**: While multiple definitions of hypergraph density exist in different contexts, we adopt a most straightforward definition as the ratio of the number of hyperedges to the number of nodes, given by Density $= \frac{|\mathcal{E}|}{|V|}$.

- **Average Hyperedge Size**: The average hyperedge size is calculated as: Average Hyperedge Size $= \frac{1}{|\mathcal{E}|} \sum_{e \in \mathcal{E}} |e|$.

- **Average Node Degree**: The average node degree represents the average number of hyperedges in which each node participates, given by: Average Node Degree $= \frac{1}{|V|} \sum_{v \in V} d(v)$.

- **Clustering Coefficient of Graph Projection**: The clustering coefficient is computed for the 2-section (or graph projection) of the hypergraph, where each hyperedge induces a clique among its nodes. This coefficient measures the likelihood that two neighbors of a node are also neighbors. It is calculated as: $C = \frac{1}{|V|} \sum_{v \in V} C(v)$, where $C(v)$ is the local clustering coefficient of node $v$ in the projected graph.

- **Modularity of Graph Projection**: The modularity of the graph projection measures the quality of a partition of the graph into communities, with higher values indicating a stronger community structure. It is given by: $Q = \frac{1}{2m} \sum_{i,j} \left[ A_{ij} - \frac{k_i k_j}{2m} \right] \delta(c_i, c_j)$, where $A_{ij}$ represents the adjacency matrix, $k_i$ and $k_j$ are the node degrees, $m$ is the total number of edges, and $\delta(c_i, c_j)$ equals 1 if nodes $i$ and $j$ belong to the same community, and 0 otherwise.

- **Modularity of Bipartite Graph Representation**: The modularity of the bipartite graph representation measures the quality of a partition of the bipartite graph, considering nodes representing original vertices and hyperedges. The formula for modularity is similar to that used for the projected graph.

## D  IMPACT OF RANK VALUE IN CELL

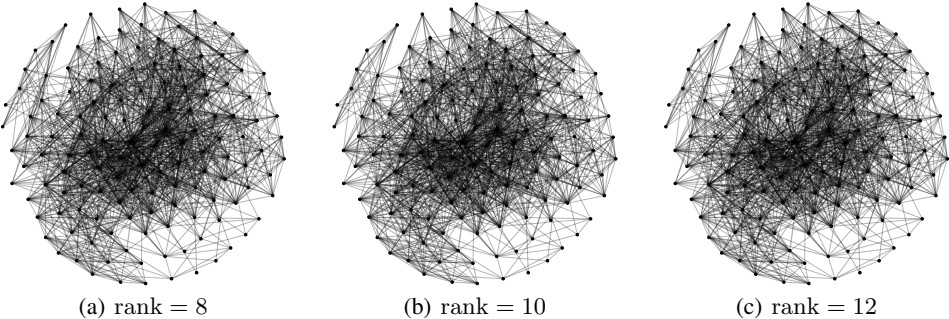

|              (a) rank = 8              |              (b) rank = 10              |              (c) rank = 12              |

Figure 4: Visualization of generated weighted graph projection of Hypergraph `email-Enron` with different rank values. Zoom in for a better view.

From fig. 4, we can conclude that, in terms of graph structure, there is almost no influence from rank value. However, rank greatly impacts the generation speed in the CELL module. High rank can accelerate the generation greatly but lose some diversity in the generated hypergraph, which is also discussed in Rendsburg et al. (2020).

## E  DIVERSITY OF THE GENERATION

To the best of our knowledge, this is an essential yet unresolved problem in the graph (and hypergraph) generation task. Below, we make the best effort to provide a fair comparison and discussion of the diversity of hypergraphs generated by our model and HyperLap. Despite the lack of discussion on diversity in hypergraph generation directly, Velikonivtsev et al. (2024) proposed a theoretical framework for evaluating graph diversity. In summary, the paper defines the diversity of a set of graphs $S$ as Diversity$(S) = F(D(G, G') : G, G' \in S)$, where $D(G, G')$ is a distance measure between two graphs, and $F$ is a function that computes the overall diversity from the set of pairwise

distances. Building on this theory, we conducted the following experiments. First, we converted the hypergraphs generated by HyperLap, HyperLap+, and HyperPLR into bipartite graph representations for each dataset[1]. In this representation, one set of vertices corresponds to the hyperedges, and the other set corresponds to the vertices connected by these hyperedges. This conversion is lossless with respect to the original hypergraph structure.

Given the large size of the graphs, classical graph distance measures like graph edit distance are NP-hard and computationally prohibitive. Instead, we followed the suggestions of Tsitsulin et al. (2018) and utilized two spectral-based distance measures introduced in : NetLSD-heat and NetLSD-wave. For the function $F$, we adopted the sum of pairwise distances between elements, as it is the most natural and widely used approach. The Table 4 presents the results of our experiments. From the results, we can see that in most cases, HyperPLR achieves graph diversity that either surpasses or is competitive to that of random generators like HyperLAP and HyperLap+. At the same time, we believe that the concept of diversity in the hypergraph generation task requires a more targeted definition and evaluation method.

Table 4: NetLSD-Heat&NetLSD-Wave results across various datasets.

| NetLSD-Heat | contact-high-school | contact-primary-school | email-Enron | email-Eu | NDC-classes |
|---|---|---|---|---|---|
| HyperLap | 0.006 | 0.002 | 0.011 | 0.015 | 0.094 |
| HyperLap+ | 0.007 | **0.056** | 0.009 | 0.007 | 0.087 |
| HyperPLR | **0.008** | 0.006 | **0.017** | **0.022** | **0.354** |
| **NetLSD-Wave** | | | | | |
| HyperLap | 0.070 | 0.033 | 0.158 | **0.262** | 1.216 |
| HyperLap+ | 0.086 | **0.975** | 0.097 | 0.127 | 0.562 |
| HyperPLR | **0.113** | 0.107 | **0.183** | 0.251 | **1.479** |

---

[1]For THERA, we encountered limitations as the provided code was not executable, and the authors only supplied a single generated sample per dataset. While we were able to compute graph metrics for this single sample, evaluating its diversity was not possible due to the absence of multiple samples.

