# OpenReview forum: "HyperPLR: Hypergraph Generation through Projection, Learning, and Reconstruction"
_ICLR.cc/2025/Conference — ICLR 2025 Poster_

### Official Review · Reviewer_WfsL · 2024-10-29

**Soundness:** 3
**Presentation:** 3
**Contribution:** 3
**Rating:** 6
**Confidence:** 4

**Summary:**

***Overview.*** The authors study the hypergraph generation task.

***Method.*** Specifically, they propose a learning-based hypergraph generative method, named HyperPLR.
HyperPLR learns the (potential) weight of each edge with GNN from the unweighted graph.
Therefore, (1) it generates an unweighted graph with CELL, and (2) it assigns edge weights by using the trained GNN.
Lastly, the weighted graph is transformed into a hypergraph.

***Experiment.*** The authors demonstrated the effectiveness of HyperPLR in several benchmark hypergraph datasets, compared to hypergraph generative baseline methods.

**Strengths:**

***S1.*** This is the first learning-based hypergraph generative method.

***S2.*** Overall method design is reasonable.

***S3.*** Various baseline methods are considered.

**Weaknesses:**

***W1 (Performance).*** Despite the large training cost, performance gain compared to the rule-based methods seems a bit marginal. Specifically, HyperLAP and THERA often outperform the proposed method.

***W2 (Scalability).*** The authors acknowledge that the proposed method lacks scalability, which is a significant limitation given that many real-world hypergraphs are typically large. Despite its slower training and generation, is the proposed method ***capable of*** generating larger hypergraphs (e.g., those with more than 10,000 nodes)?

***W3 (Independent edge generation).*** Many real-world hypergraphs include timestamps, as interactions occur over time. Furthermore, interactions may influence one another, suggesting that hyperedges may not be independent. However, to my understanding, the proposed method generates pairwise edges without accounting for temporal aspects and dependencies.

Please clarify if my understanding or claims are wrong.

**Questions:**

See the mentioned weaknesses.

---

> ### Author Response · Authors · 2024-11-24
>
> Thanks for your insightful comments and for acknowledging novelty and model design. Below, we respond to your specific comments.
>
> **W1:**\
> We thank the reviewer for their observation. Below, we address the concerns regarding training cost and performance comparison with rule-based methods：
> 1. The training cost for HyperPLR is not as large as implied. Taking the email-Eu dataset (998 nodes and 234,760 hyperedges) as an example: Node embedding learning takes only 4.5 seconds. CELL and GNN training together take 20 seconds, which is a **one-time cost**. Once trained, these components can be reused for similar datasets or applications.
> 2. There is a trade-off between graph statistics similarity and generation diversity: while methods like HyperLAP and THERA often optimize for closer graph statistics to the original hypergraph, our approach **explicitly controls training iterations to ensure diversity** in the generated hypergraphs. This is intentional, as the exact reproduction of graph statistics may lead to overfitting and reduced generalizability in practical scenarios. HyperPLR aims to **strike a balance between structural fidelity and diversity** in generated hypergraphs, which is critical for applications requiring realistic variations of the original data.
>
> **W2:**\
> We thank the reviewer for raising concerns regarding scalability and the capability of HyperPLR to generate larger hypergraphs. Below, we clarify the limitations and capabilities of our method:
> 1. The training and generation phases of HyperPLR are efficient compared to many learning-based graph generation methods. For instance: Node embedding learning, CELL training, and the GNN weight predictor collectively incur a manageable one-time cost, as demonstrated on the email-Eu dataset (998 nodes and 234,760 hyperedges) with a total training time of ~25 seconds.
> 2. The scalability bottleneck is mainly in clique finding, which occurs before the reconstruction phase. This step is sensitive to the density of the projection graph, rather than the number of nodes or hyperedges directly. For sparse projection graphs, clique enumeration remains computationally manageable, even for hypergraphs with over 10,000 nodes. For dense projection graphs, the computational cost increases significantly, similar to other graph reconstruction methods that involve clique-based or edge-cover algorithms, such as SHyRe [1].
>
> **W3:**\
> We appreciate the reviewer’s observation about the consideration for temporal aspects and dependencies. Here, we clarify the scope and focus of our proposed method. Hypergraph generators can be broadly categorized into **static and dynamic models**. **HyperPLR is a static generator**, designed to model the global structural properties of a hypergraph without explicitly incorporating temporal dynamics or interdependencies between hyperedges. This focus ensures broad applicability to static datasets and scenarios where temporal information is unavailable or secondary. While real-world hypergraphs often include temporal interactions, accounting for these dynamics requires a fundamentally different framework (e.g., incorporating temporal evolution models or causal dependencies). This is beyond the scope of HyperPLR, which aims to provide a static yet robust and general-purpose framework for hypergraph generation.
>
>
> [1] From Graphs to Hypergraphs: Hypergraph Projection and its Reconstruction. ICLR 2024

---

> > ### Comment · Reviewer_WfsL · 2024-11-25
> > **Thank you for the responses**
> >
> > Dear Authors
> >
> > Thank you for your responses.
> > I understand the responses regarding W2 and W3, while I have an additional question on your responses to my W1.
> >
> > I agree that the fidelity and diversity are in trade-off, and balancing these two factors is important for generative models.
> > In addition, I think the similarity between real-world hypergraphs and synthesis can be a ***fidelity measure***.
> > Then, how can we evaluate ***diversity*** of hypergraph generators?
> > To my knowledge, both HyperLAP and THERA are “random generators”, which can generate diverse hypergraphs, capable of having diversity.
> >
> > Is there any empirical comparions or theoretical guarantee that the proposed method has higher “diversity” compared to HyperLAP and/or THERA?

---

> ### Author Response · Authors · 2024-11-27
> **Thank you for your timely responses**
>
> We thank the reviewer for highlighting the importance of balancing fidelity and diversity in generative models. To the best of our knowledge, this is still an essential yet unresolved problem in the graph (and hypergraph) generation task. Below, we make the best effort to provide a fair comparison and discussion of the diversity of hypergraphs generated by our model and HyperLap (see note for THERA).
>
> Previously, we empirically controlled the edge overlapping parameter in the CELL model [1] of HyperPLR to be less than or equal to 0.8 (and ≤ 0.6 for large graphs like Email-Eu). This limitation ensured that the topology generated in the HyperPLR did not become overly similar to the original hypergraph's projection, thereby maintaining the diversity of the generated samples.
>
> During this period, we conducted some literature reviews related to the graph generation task. Despite the lack of discussion on diversity in hypergraph generation directly, we discovered a paper at this year’s NeurIPS [2] that proposed a theoretical framework for evaluating graph diversity. In summary, the paper defines the diversity of a set of graphs $S$ as $\text{Diversity}(S) = F(\{D(G, G^\prime): G, G^\prime \in S\})$, where $D(G, G^\prime)$ is a distance measure between two graphs, and $F$ is a function that computes the overall diversity from the set of pairwise distances.
>
> Building on this theory, we conducted the following experiments, we have updated the evaluation code, and data results in the [anonymous repository](https://anonymous.4open.science/r/hyperplr-C0F3/diversity.ipynb). First, we converted the hypergraphs generated by **HyperLap**, **HyperLap+**, and **HyperPLR** into bipartite graph representations for each dataset. In this representation, one set of vertices corresponds to the hyperedges, and the other set corresponds to the vertices connected by these hyperedges. This conversion is lossless with respect to the original hypergraph structure.
>
> Given the large size of the graphs, classical graph distance measures like graph edit distance are NP-hard and computationally prohibitive. Instead, we followed the suggestions of paper [2] and utilized two spectral-based distance measures introduced in paper [3]: **NetLSD-heat** and **NetLSD-wave**. For the function  **$F$**, we adopted the sum of pairwise distances between elements, as it is the most natural and widely used approach. The tables below present the results of our experiments:
>
> (NetLSD-heat)|contact-high-school | contact-primary-school | email-Enron | email-Eu | NDC-classes
> --- | --- | --- | --- | --- | ---
> HyperLap | 0.006 | 0.002 | 0.011 | 0.015 | 0.094
> HyperLap+ | 0.007 | **0.056** | 0.009 | 0.007 | 0.087
> HyperPLR |**0.008** | 0.006 | **0.017** | **0.022** | **0.354**
>
> (NetLSD-wave) | contact-high-school | contact-primary-school | email-Enron | email-Eu | NDC-classes
> --- | --- | --- | --- | --- | ---
> Hyperlap | 0.070 | 0.033 | 0.158 | **0.262** | 1.216
> Hyperlap+ | 0.086 | **0.975** | 0.097 | 0.127 | 0.562
> HyperPLR | **0.113** | 0.107 | **0.183** | 0.251 | **1.479**
>
> From the results, we can see that in most cases, HyperPLR achieves graph diversity that either surpasses or is competitive to that of random generators like HyperLAP and HyperLap+. We hope this can demonstrate the diversity achieved by HyperPLR during generation.
>
> At the same time, we believe that the concept of diversity in the hypergraph generation task requires a more targeted definition and evaluation method. We plan to conduct further research on this in future work and warmly welcome any further discussions with you on this topic.
>
> ------
> [1] NetGAN without GAN: From Random Walks to Low-Rank Approximations. ICML 2020.\
> [2] Challenges of Generating Structurally Diverse Graphs. NeurIPS 2024.\
> [3] NetLSD: hearing the shape of a graph. KDD 2018.
>
> **Note**: For THERA, we encountered limitations as the provided code was not executable, and the authors only supplied a single generated sample per dataset. While we were able to compute graph metrics for this single sample, evaluating its diversity was not possible due to the absence of multiple samples.

---

> > ### Comment · Reviewer_WfsL · 2024-11-27
> > **Thank you for the response**
> >
> > Thank you for your follow-up responses.
> >
> > The results are convincing, and now I agree with the authors' claim.
> >
> > While the proposed method has several limitations (e.g., scalability issues), I think the proposed method has a contribution as the first learning-based hypergraph generator, which can serve as a good starting point for this field.
> >
> > In light of this, I would like to increase my score $5 \rightarrow 6$.

---

> ### Author Response · Authors · 2024-11-28
>
> Thank you for taking the time to review our work and for providing thoughtful feedback. We greatly appreciate your recognition of our efforts and are encouraged by your positive response. Your constructive comments have been invaluable in helping us refine the paper, and we remain dedicated to further enhancing its quality.

---

### Official Review · Reviewer_C2HN · 2024-10-31

**Soundness:** 2
**Presentation:** 2
**Contribution:** 2
**Rating:** 6
**Confidence:** 4

**Summary:**

This paper introduces HyperPLR, a hypergraph generative model employing a three-stage pipeline: projection, learning, and reconstruction. HyperPLR first projects the input hypergraph to a weighted projected graph and then utilizes this structure to learn graph structures, including both adjacency and edge weights. Then it reconstructs new hypergraphs by addressing a clique cover problem on the generated weighted projected graph. The authors evaluated HyperPLR by comparing it against SOTA hypergraph generative methods.

**Strengths:**

- HyperPLR introduces a new approach to hypergraph generation, based on three-stage framework.
- The authors thoroughly discuss the limitations of the method.
- Code and datasets are available.

**Weaknesses:**

**Clarity and presentation issues**
- Important definitions, such as "node degree in a weighted graph," are missing. This makes terms like "the degree of each node corresponds to its total frequency of appearance in hyperedges" unclear.
- Inconsistent notations (e.g., Graph A in Figure 2 and Graph G in the text) are used.
- "G coefficient/modularity" and "B modularity" in Table 1 lack proper definitions.
- It is unclear in line 202, why weighted projection offers a more "compact representation" compared to the bipartite graph representation of hypergraphs.
- "Strong global structures" in line 214 require better contextual discussions.
- A typo in line 138: $(v_i, v_j)\in e_i$.

**Method descriptions**
- While the authors suggest the "novelty" of the weighted projection of hypergraphs, this approach is well-studied and employed in prior hypergraph studies, even in early work like "Learning with Hypergraphs: Clustering, Classification, and Embedding" (2006).
- More details are needed about the edge weight prediction GCN module. For example, how is the loss function designed? - Does it require any negative samples? How is the bias in the edge weight distribution addressed?
- What are the bottlenecks in HyperPLR? What is its space/time complexity?

**Evaluation**
- The model's performance is not impressive enough; for example, it is outperformed by HyperLap in many cases. While the authors argue HyperPLR does not require structural parameters directly from the input, non-trivial features such as number of nodes, edge weights, and maximum hyperedges are given.
- More advanced evaluation metrics, such as those used in baseline papers, could have been considered.
- Table 2 lacks a comparative baseline.
- The effectiveness of key components in HyperPLR remains unclear. For example, how does HyperPLR perform when excluding CELL (i.e., using the input graph G directly)? Is the proposed GWC module effective?


**In summary**, the paper has room for improving the clarity of definitions and consistency of notation, highlighting the novelty, providing a more thorough description of the method, and strengthening the experiments with more comprehensive evaluation metrics and ablation studies.

**Questions:**

Please refer to the weaknesses.

---

> ### Author Response · Authors · 2024-11-24
>
> Thanks for your insightful comments. Below we respond to the specific comments:
>
> **W1**\
> We thank the reviewer for highlighting these concerns and provide the following clarifications and resolutions. Below, we provide detailed responses and outline how these issues will be addressed:
> - **Definitions:** We will include precise definitions in the Preliminaries section. For example, “Node degree in a weighted graph is the sum of the weights of all edges connected to the node.” Additionally, terms like “frequency of appearance in hyperedges” will be clarified to ensure clear links between hypergraphs and weighted graph projections.
> - **Notation Consistency:** We will standardize all notations, ensuring consistency between figures and the text.
> - **Metrics in Table 1:** Definitions for “G coefficient/modularity” and “B modularity” will be added under the evaluation metrics, explaining that they refer to clustering coefficients/modularity in graph and bipartite representations, respectively.
> - **Compact Representation:** The statement in line 202 will be expanded to explain that weighted projections condense hyperedge information into edges, reducing graph size compared to bipartite representations, which require separate hyperedge nodes. An example or diagram will also be added.
> - **Strong Global Structures:** We will refine “strong global structures” in line 214 to refer to long-range dependencies like overlapping cliques or communities in large hypergraphs, adding an example for clarity.
> - **Typo in Line 138:** The typo  $(v_i, v_j) \in e_i$  will be corrected.
> We hope these revisions will enhance the clarity and presentation of the paper and address the reviewer’s concerns effectively.
>
> **W2**
> 1. We thank the reviewer for pointing out that weighted projections of hypergraphs have been explored in prior work. We would like to clarify that we do not claim the novelty of using weighted graph projections as a standalone contribution. Instead, the novelty of our work lies in the proposed HyperPLR framework and the Weighted Clique Edge Cover (WCEC) formulation, which leverage weighted projections to address critical gaps in hypergraph generation and reconstruction. While weighted projections are indeed well-established, the HyperPLR framework integrates this step with a learning phase that employs GNN-based edge weight prediction to refine the weighted graph. What's more, the HyperPLR framework integrates a reconstruction phase that introduces the WCEC problem and the Greedy Weighted Cover (GWC) algorithm for reconstructing hypergraphs. This combination enables the effective preservation of higher-order relationships and overlapping hyperedges, which is beyond the focus of prior works.
> 2. We appreciate the reviewer’s interest in the details of our edge weight prediction GCN module. Below, we provide clarification:\
> The edge weight prediction is formulated as a regression task, where the GCN predicts edge weights based on node embeddings and the graph structure. The loss function is designed as the Mean Squared Error (MSE) between the predicted edge weights and the ground truth weights from the original weighted projection graph. Specifically, the loss function is given as:  $\text{Loss} = \sum_{(i,j) \in E} \big( w^\prime_{i,j} - w_{i,j} \big)^2$ where $w^\prime_{i,j}$ is the predicted weight for edge $(i, j)$, and $w_{i,j}$ is the ground truth weight. Since this is a regression task and not a classification problem, the GCN does not require negative samples (i.e., edges that do not exist). Instead, the focus is on minimizing the error between the predicted and actual weights for existing edges. The GCN learns to approximate the edge weights based on node features and adjacency relationships within the weighted graph projection.\
> In most cases, the edge weight distribution predicted by the GNN model closely matches the true edge weight distribution of the weighted projection. Both distributions appear to resemble exponential distributions or other heavily right-skewed distributions, where a significant portion of the data is concentrated near smaller values, with frequencies decreasing as the values increase. [This picture](https://anonymous.4open.science/r/hyperplr-C0F3/GNN_bias.png) in the anonymous repository is the visualization of their distributions for the email-Eu dataset.
> 3. We thank the reviewer for raising this important question. The primary bottleneck in HyperPLR lies in the enumeration step of potential cliques during the Weighted Clique Edge Cover (WCEC) reconstruction phase. This step, while crucial for generating candidate cliques, becomes computationally expensive as the size of the graph and the number of hyperedges increase. In practice, we don’t typically witness such an explosion when enumerate the cliques. The space/time complexity can be found in the general response.

---

> ### Author Response · Authors · 2024-11-24
>
> **W3:**
> 1. The experimental results in our paper reflect a comprehensive evaluation across multiple datasets and metrics (e.g., density, average degree, clustering coefficient, modularity, etc.), rather than selecting specific metrics or datasets where HyperPLR excels. While HyperPLR may not outperform methods like HyperLap on all metrics, it achieves competitive and consistent performance across a wide range of datasets and metrics, demonstrating its generality and robustness. Our goal is not solely to achieve state-of-the-art performance but to offer a promising framework for hypergraph generation that balances simplicity and effectiveness. HyperPLR integrates weighted graph projection, GNN-based weight prediction, and WCEC reconstruction in a modular and efficient manner.\
> Unlike parameter-specific methods like HyperLap and HyperDK, which directly require detailed structural inputs (e.g., hyperedge size distributions or specific node degree properties), HyperPLR only utilizes basic input parameters such as the number of nodes, edge weights, and maximum hyperedge size. These are general properties available in most real-world hypergraph datasets and do not encode problem-specific structural assumptions. This makes HyperPLR more adaptable and less reliant on dataset-specific tuning.
> 2. We appreciate the reviewer’s suggestion to consider more advanced evaluation metrics. Below, we explain our rationale for selecting the metrics used in the paper:
>     - As noted in the literature, there is no universally accepted set of evaluation metrics for hypergraph generation. Most baseline methods propose their own tailored metrics to emphasize the strengths of their specific approaches, but these metrics are inherently biased and not widely adopted by the research community.
>     -  To ensure fair and unbiased comparisons, we used the most commonly accepted graph metrics, including density, average node degree, clustering coefficient, and modularity. These metrics are widely recognized for evaluating the structural fidelity and quality of graph representations and provide a general assessment of hypergraph generation performance.
>     - By focusing on standard graph metrics, we aimed to evaluate HyperPLR’s ability to generate hypergraphs that are structurally consistent with real-world datasets, independent of any specific method’s tailored evaluation framework.
>
>     To ensure a fair and transparent evaluation, we adopted commonly accepted graph metrics rather than method-specific tailored metrics. These metrics provide a robust basis for comparing HyperPLR with existing methods without introducing bias. We will clarify this rationale in the revised manuscript.
> 3. We appreciate the reviewer’s observation regarding the absence of a comparative baseline in Table 2. We need to clarify the purpose of this table and why a direct comparison with other methods is not applicable. Table 2 is designed to illustrate the evolution of graph metrics across the three stages of the HyperPLR generation process: original graph $G_w
> $, learned graph $G_w^L$, re-projected graph obtained through GWC $G_w^R$. The goal is to demonstrate that HyperPLR effectively preserves structural properties throughout the generation process while introducing a desirable level of diversity in the reprojected graph.
> 4. We appreciate the reviewer’s suggestion to clarify the effectiveness of key components in HyperPLR. Below, we address the two specific points raised.
>     - **Effectiveness of CELL.** The inclusion of CELL in HyperPLR is critical for generating a low-rank transition matrix that incorporates spectral information from the input graph. This process captures important global properties, enabling the GNN-based weight predictor to work on a refined graph topology rather than directly on the original projection graph. If we use the input graph directly, the generated graph would lack diversity, leading to poorer generalization and reconstruction fidelity.
>     - **Effectiveness of the GWC.** Empirical evidence (e.g., Table 2) demonstrates that GWC achieves prominent reconstruction ability between the learned graph $G_w^L$ and reconstructed weighted graphs $G_w^R$. This indicates that GWC effectively captures the frequency-based relationships encoded in the edge weights.

---

> > ### Comment · Reviewer_C2HN · 2024-11-26
> >
> > Thank you for the clarifications. Some of my concerns have been addressed, and I have adjusted my evaluation accordingly. Below are my follow-up comments:
> >
> > - Regarding W2-1, as the authors noted, the weighted graph projection itself is not a novel contribution of this work. Thus, the statement in line 140 "we adopt a novel graph projection method for hypergraphs - weighted graph projection" should be rephrased to avoid overstatement.
> > - Regarding W3-1, while the authors argue that HyperPLR uses edge weight information as a *basic* property, these weights may capture non-trivial structural information and are only utilized in HyperPLR.
> > - Regarding W3-2, while hypergraph generation is indeed less explored compared to graph generation, other advanced evaluation measures are available. For example, please refer to Appendix F.2.2. of [1]. I believe the current evaluation metrics used in Table 1 are not comprehensive enough to capture the structural properties of hypergraphs.
> >
> >
> > [1] From Graphs to Hypergraphs: Hypergraph Projection and its Remediation (ICLR 2024).

---

> ### Author Response · Authors · 2024-11-29
> **Thank you for the response**
>
> We are pleased that our response was able to address your concerns, and we sincerely appreciate your willingness to re-evaluate our work and the thoughtful follow-up comments. Your questions and suggestions are instrumental in further strengthening our paper. Below are our responses to the remaining comments:
>
> **W2-1:**\
> We agree with the reviewer that the weighted graph projection itself is not a novel contribution of this work. What we aim to convey is that this is the first time weighted graph projection has been utilized in learning-based hypergraph generation. We will rephrase the statement in line 140 to avoid overstating the novelty of the weighted projection in our paper. The adjustment will ensure clarity and avoid any misinterpretation of our claims.
>
> **W2-2:**\
> We appreciate the reviewer’s observation that the edge weight used in HyperPLR may capture non-trivial structural information and acknowledge that these edge weights are uniquely utilized in our framework. We argue that **edge weights are more accessible and inherently more robust in practice**, especially in scenarios where hypergraph-level structural information is either unavailable or unpublished. These scenarios inspired paper [1], which in turn strongly influenced our work. Below, we elaborate on this point:
> - In many real-world cases, **hypergraph data is not directly available**, as data collection methods often capture only pairwise interactions rather than higher-order relationships. For example, the social science studies often record pairwise interactions using physical proximity sensors or communication logs but lack direct records of multi-person interactions [2]. However, the frequency of these pairwise interactions can be used to approximate edge weights for graph projection.
> - Even when hypergraph-level data exists, it is often **unpublished or inaccessible** in widely used datasets. For instance, the benchmark such as ogbn-proteins [3] provide weighted graph based data  but not their corresponding hypergraph structures. In such cases, weighted graph projection offers a practical and robust intermediate representation.
>
> Unlike parameter-based hypergraph generators, HyperPLR effectively leverages edge weights without requiring detailed global hypergraph parameters, which are difficult to obtain when the hypergraph is unobservable or unpublished. This makes HyperPLR both practical and widely applicable. We will incorporate these clarifications into the revised manuscript.

---

> ### Author Response · Authors · 2024-11-29
>
> **W3-2:**\
> We acknowledge the reviewer’s suggestion to expand the evaluation metrics to the Appendix F.2.2. of paper [1]. Below are our experimental results. Notably, the simplicial closure metric from paper [1], which evaluates the temporal evolution process of dynamic hypergraphs, is not applicable to our work or the baseline static generation models. Therefore, we use a similar concept, triangle numbers, which is a widely used graph structural metric [4]. For density, we have already presented the results in Table 1, so it is not repeated here.
>
> Similar to the approach in paper [1], we standardized the metrics for triangle numbers and diameter using the formula: $\frac{|x_1 - x_2|}{x_1}$, where  $x_1$  and  $x_2$ represent the values of the ground truth and the generated graphs, respectively. For degree distribution and singular value distribution, we compared their cross-entropy.  **A smaller value indicates better alignment for all the metrics**. Also, we have updated all evaluation codes and data in the paper’s [anonymous repository](https://anonymous.4open.science/r/hyperplr-C0F3/advanced_property.ipynb).
>
> From the results, we observe that HyperPLR outperforms or is competitive with other models across most datasets and metrics. We hope this demonstrates that we did not engage in any cherry-picking metrics when presenting the results in our original paper. Furthermore, as [discussed with Reviewer WfsL](https://openreview.net/forum?id=TYnne6Pa35#:~:text=NetLSD), we have shown that HyperPLR generates graphs with higher diversity compared to other models while maintaining strong structural similarity to the original data.
>
> (diameter) | contact-high-school | contact-primary-school | email-Enron | email-Eu | NDC-classes
> --- | --- | --- | --- | --- | ---
> $\text{HyperDK}_{00}$ | 0.500 | 0.333 | 0.500 | 0.667 | 0.778
> $\text{HyperDK}_{11}$ | 0.250 | 0.333 | 0.250 | 0.333 | 0.533
> Hyperlap | **0.200** | 0.067 | 0.250 | 0.333 | 0.422
> Hyperlap+ | 0.250 | 0.333 | 0.050 | 0.133 | 0.289
> TheRA | 0.250 | **0.000** | 0.250 | 0.667 | 0.222
> HyperPLR | 0.250 | **0.000** | **0.000** | **0.033** | **0.000**
>
> (triangles_num) | contact-high-school | contact-primary-school | email-Enron | email-Eu | NDC-classes
> --- | --- | --- | --- | --- | ---
> $\text{HyperDK}_{00}$ | 167.117 | 21.456 | 41.427 | 156.738 | 184.785
> $\text{HyperDK}_{11}$ | 84.440 | 15.988 | 10.867 | 41.227 | 79.077
> Hyperlap | 0.599 | 0.409 | 3.315 | 8.998 | 3.536
> Hyperlap+ | 0.942 | 0.284 | 1.517 | 2.981 | 2.124
> TheRA | 0.468 | **0.198** | 2.451 | 3.725 | 1.638
> HyperPLR | **0.316** | 0.354 | **0.055** | **0.742** | **0.281**
>
> (degree_distribution) | contact-high-school | contact-primary-school | email-Enron | email-Eu | NDC-classes
> --- | --- | --- | --- | --- | ---
> $\text{HyperDK}_{00}$ | 23.026 | 23.026 | 23.026 | 23.026 | 23.026
> $\text{HyperDK}_{11}$ | 21.002 | 22.635 | 12.406 | 8.490 | **3.960**
> Hyperlap | 5.958 | 8.072 | 9.294 | 6.946 | 4.415
> Hyperlap+ | 5.336 | 8.812 | 9.023 | **6.448** | 4.474
> TheRA | 6.634 | 10.530 | 7.986 | 15.315 | 14.325
> HyperPLR | **5.065** | **7.657** | **4.639** | 7.424 | 4.020
>
> (singular_value_distribution) | contact-high-school | contact-primary-school | email-Enron | email-Eu | NDC-classes
> --- | --- | --- | --- | --- | ---
> $\text{HyperDK}_{00}$ | 6.212 | 5.730 | 4.653 | 6.422 | 6.750
> $\text{HyperDK}_{11}$ | 5.462 | 5.147 | 4.513 | 6.345 | 7.008
> Hyperlap | 5.369 | 5.043 | 4.470 | 6.293 | 6.589
> Hyperlap+ | 5.372 | 5.056 | 4.520 | 6.319 | 6.560
> TheRA | 5.377 | 5.054 | 4.465 | 6.339 | 6.620
> HyperPLR | **5.347** | **5.032** | **4.451** | **6.278** | **6.432**
>
> Thank you for your insightful advice and for providing us with the opportunity to engage in this valuable discussion. We hope our response satisfactorily addresses your concerns, and we will incorporate all these meaningful empirical results and discussions in the final version of our paper. We remain open and willing to address any further concerns you may have.
>
> ---
> [1] From Graphs to Hypergraphs: Hypergraph Projection and its Remediation (ICLR 2024).\
> [2] Using wearable proximity sensors to characterize social contact patterns in a village of rural malawi (EPJ Data Science 2021).\
> [3] Open Graph Benchmark: Datasets for Machine Learning on Graphs (NeurIPS 2020).\
> [4] Implications of sparsity and high triangle density for graph representationlearning (AISTATS 2023).

---

> > ### Comment · Reviewer_C2HN · 2024-12-02
> >
> > Thank you for the response and for conducting additional experiments to address my concerns. I have adjusted my score and look forward to reviewing the updated paper.

---

> > > ### Author Response · Authors · 2024-12-02
> > >
> > > Thank you for reviewing our paper and providing valuable feedback that helps consistently improve the paper quality. We are glad that our response can address your concerns as well. We will reorganize our submission in the next version according to your suggestions. Once again, thank you for the significant effort from you during the reviewing process.

---

### Official Review · Reviewer_ZxbQ · 2024-11-03

**Soundness:** 2
**Presentation:** 2
**Contribution:** 3
**Rating:** 6
**Confidence:** 3

**Summary:**

Hypergraphs are essential in modeling high-order relationships. Hypergraph generative models are crucial for promoting and validating the understanding of these structures. In this paper, the authors introduce a novel hypergraph generative approach, HyperPLR, encompassing three phases: Projection, Learning, and Reconstruction. They have evaluated HyperPLR on existing real-world hypergraph datasets, demonstrating superior performance and validating the effectiveness of HyperPLR.

**Strengths:**

1. Hypergraph generation is important for understanding the nature of high-order relationships.
2. The proposed method is based on a simple, intuitive idea, yet shows good empirical performance on various datasets.
3. Compared to deep learning frameworks, the proposed method is easier to interpret and trust.

**Weaknesses:**

1. Technical novelty is limited. (a) The idea to create a hypergraph structure from an ordinary (clique-expanded) graph has been studied already in (Bresler et al., 2024). (b) Training a GNN-based weight predictor for weighted graph structures seems new, but the technical contribution is limited, as the authors use typical GCN with node2vec-based features. (c) The proposed algorithm for weight cover is also a simple greedy algorithm which has no theoretical guarantee on its performance.
2. Writing quality should be improved in general. (a) Section 4 includes things that are part of HyperPLR, e.g., lines 189 - 192, 233 - 243, etc. It would be easier to read the paper if the authors separate them from the “proposed” ones. (b) Some knowledge about MCEC should be given so that readers can understand the propositions in a better context.
3. Theoretical claims do not support the proposed method well. Lemma 2 is straightforward, and it does not always support that WCEC is better than MCEC, considering that acquiring good edge weights is difficult. Even with the GNN predictor, we cannot guarantee the quality of this solution.

**Questions:**

1. Refer to the weaknesses above.
2. Can we replace CELL with any recent graph generator models, like those introduced in [1]?

- [1] Liu et al. “Generative Diffusion Models on Graphs: Methods and Applications.” IJCAI 2023

---

> ### Author Response · Authors · 2024-11-24
>
> Thanks for your valuable comment, and nice suggestions, as well as for acknowledging our non-trivial contributions. Below, we respond to your specific comments.
>
> **W1:**\
> HyperPLR is inspired by some excellent previous works. However, we must note that the technical contributions of HyperPLR are highly non-trivial.
> 1. Unlike [Bresler et al., 2024], our approach integrates the frequency of node co-occurrences into the weighted projection. This provides a more nuanced representation of hypergraph relationships, capturing higher-order structural information often lost in simpler clique-expanded methods. Our proposed WCEC problem, extends the reconstruction process by explicitly considering weights in clique covers, a challenging direction unexplored by prior work. This additional layer of detail enables the generation of hypergraphs that better match the properties of the original datasets, as demonstrated by our experiments.
> 2. While we employ established techniques such as GCNs and node2vec embeddings, our novelty lies in the introduction of the principled HyperPLR framework. Although we employed some previous works at the module-level, the effective and principled design at framework-level is non-trivial. Specifically, the combination of CELL-generated topology with GNN-based weight prediction and the innovative WCEC formulation provides a three-stage learning mechanism that effectively models both graph structure and edge weights simultaneously. For more concerns about the novelty,  please kindly refer to the general response for the novelty claim.
> 3. For the theoretical guarantee, please kindly refer to the general response for the theoretical guarantee claim.
>
> **W2:**\
> We appreciate the reviewer’s constructive feedback regarding the clarity and organization of the manuscript. Below, we outline how we plan to address the issues raised to improve the readability and comprehension of our work:
> 1. To improve clarity, we will restructure Section 4 by separating general background content into the preliminaries section and focusing another section exclusively on the Proposed HyperPLR Framework. This distinction will help readers easily identify our contributions.
> 2. We will add a brief introduction to Minimum Clique Edge Cover (MCEC) in Section 3 to explain its purpose, limitations, and how it connects to our Weighted Clique Edge Cover (WCEC) approach. This will provide the necessary context for understanding the propositions.
>
> **W3:**
> 1. We agree with the reviewer that Lemma 2 is relatively straightforward, as it builds on the definitions of MCEC and WCEC. However, its purpose is not to claim the theoretical superiority of WCEC over MCEC but to provide an interpretive bridge between these two approaches. WCEC incorporates edge weights into the reconstruction process, which encodes the frequency of co-occurrence between nodes in hyperedges. This allows WCEC to capture both the structure of the hypergraph and the strength of relationships between nodes. In contrast, MCEC focuses solely on minimizing the number of cliques, ignoring this frequency information, which limits its ability to reconstruct overlapping or nested hyperedges accurately. Also, solving MCEC to reconstruct the hypergraph cannot guarantee the hypergraph size, which empirically tends to generate a significantly smaller number of hyperedges compared to the original hypergraph [1].
> 2. We acknowledge the reviewer’s concern that our method lacks strong theoretical guarantees. We provide the theoretical guarantee claim in the general response.
>
> **Q2:**\
> our novelty mainly lies in the framework/pipeline and WCEC. While CELL has specific advantages, we acknowledge that recent advances in graph generation may offer improvements in flexibility, scalability, or modeling power. Replacing CELL with a newer model is indeed feasible within our framework because: 1) The HyperPLR framework is modular, allowing for the substitution of the graph generation step with any model that can generate realistic weighted graph structures. 2) The weighted graph generator only needs to output an adjacency matrix and weights that are compatible with the downstream reconstruction process.
>
> CELL was chosen as the graph generation component in HyperPLR primarily due to its: 1)Spectral Efficiency: CELL focuses on spectral properties, which are computationally efficient to compute and scale well for large graphs; 2) Alignment with Weighted Graph Projection: CELL provides a low-rank transition matrix, which integrates seamlessly with our pipeline, particularly for predicting edge weights and reconstructing hypergraphs.
>
> [1] From Graphs to Hypergraphs: Hypergraph Projection and its Reconstruction. ICLR 2024

---

> > ### Comment · Reviewer_ZxbQ · 2024-11-25
> >
> > I appreciate the response. While I’m still uncertain whether this work makes significant technical contributions, I believe it is based on solid ideas and presents reasonable outcomes. I have adjusted my score to a 6, though I am not strongly confident in my opinion.

---

> > > ### Author Response · Authors · 2024-11-25
> > >
> > > Thank you for recognizing our efforts and for raising the rating. We are delighted that you found our rebuttal satisfactory and appreciate your valuable feedback. We are committed to further improving the paper and ensuring it makes a meaningful contribution to the research community.

---

### Author Response · Authors · 2024-11-24
**General Response**

We sincerely appreciate the reviewers’ time, valuable feedback, and constructive suggestions.  Your expertise and insights have been instrumental in improving the quality and clarity of our work. The primary concerns revolve around the novelty of the WCEC and the theoretical guarantee for GWC. Here, we provide a consolidated response in the general response section.

1. **Novelty Claim.**
We acknowledge that weighted projections of hypergraphs have been explored in prior work. However, the novelty of our work does not lie in the use of weighted projections as a standalone contribution, but in the principled and effective integration of components into the HyperPLR framework:
    - **Topology Generation:** HyperPLR uses CELL to generate a low-rank topology that serves as a robust foundation for weight prediction.
    - **Edge Weight Prediction:** A GNN-based model refines the weighted graph using learned embeddings, effectively bridging graph structure and edge weights.
    - **Reconstruction via WCEC:** We introduce the Weighted Clique Edge Cover (WCEC) problem and the Greedy Weighted Cover (GWC) algorithm, explicitly considering edge weights to enable the preservation of higher-order relationships and overlapping hyperedges. This challenging direction has not been explored in prior work.

    The HyperPLR framework combines these steps into a three-stage learning mechanism that models both graph structure and edge weights simultaneously. This integration allows us to reconstruct hypergraphs that better match the properties of the original datasets, as demonstrated by our experiments. The novelty of HyperPLR lies in its framework-level integration of established and novel components, rather than in isolated techniques such as weighted projections or GNNs. We hope this clarifies our contributions and addresses concerns about the novelty of our work. Additionally, to the best of our knowledge, we are the only learning-based hypergraph generation paper that has been open-sourced, where the source code can be found at the anonymous repository in [here](https://anonymous.4open.science/r/hyperplr-C0F3/README.md).
2. **Theoretical guarantee for GWC.**
Reviewer ZxbQ, C2HN raised concerns about our theoretical guarantee of performance and complexity for GWC. Here, we provide some clarifications from the optimization perspective and the computational complexity perspective.
    - **From an optimization perspective:** for the MCEC problem, which aims to select the minimum number of cliques to achieve edge coverage, can be transformed into a set cover problem. This allows for a theoretical analysis of its greedy algorithm, ultimately proving that the greedy approach achieves a 2-approximation of the optimal solution. In contrast, WCEC optimizes the L1 distance between the coverage counts of all edges and their corresponding weights, inherently making it an integer programming problem. This non-standard objective function poses significant challenges for all heuristic algorithms to achieve effective optimization guarantees on general graphs. In practice, greedy algorithms often yield good solutions for similar optimization problems, even if a formal approximation ratio isn’t established.
    - **From a computational complexity perspective:** without specific properties of the graph or additional constraints, it will be challenging to establish a precise analysis for the computation complexity. Therefore, we assume the weights in the weighted projection graph are in uniform distribution. Let $n$ be the node number, $m$ be the hyperedge number, and $S$ be the  grand sum of the weighted matrice $W$, each edge has the probability $p$ to be covered, where $p = 1 - (\frac {\binom{n}{2} - 1}{\binom{n}{2}})^S$. Let $Z_d$ be a random variable corresponding to the number of cliques with size $d$. We have $\mathbb{E}(Z_d) = \binom{n}{d}p^{\binom{d}{2}}$. Let $Z$ be the clique number, so we have $Z = \sum_{d=2}^n \mathbb{E}(Z_d) = \sum_{d=2}^n\binom{n}{d}p^{\binom{d}{2}}\leq (1+p)^n$. The time complexity of the algorithm is $O(m (2 - (\frac {\binom{n}{2} - 1}{\binom{n}{2}})^S)^n)$.

---

### Author Response · Authors · 2024-12-02
**Summary of Reviews and Discussion**

As the discussion period comes to an end, we would like to express our gratitude to all the reviewers for their constructive discussions and positive feedback. Based on these suggestions, we conducted additional theoretical and empirical analyses, incorporating numerous experiments to further demonstrate our method’s effectiveness in achieving both **similarity** and **diversity** in hypergraph generation. These contributions have significantly improved the quality of our paper.

We would like to summarize the key takeaways and contributions highlighted during the discussion. **Learning-based hypergraph generation** remains a challenging yet vital problem in the field. The difficulties stem not only from modeling and learning from large-scale hypergraph data but also from defining and evaluating **similarity** and **diversity** in generated hypergraphs. In our paper, we made an innovative attempt to tackle these challenges. We decomposed the hypergraph generation process into three modular steps: **projection, learning, and reconstruction**. This decomposition alleviates the computational burden on the learning module and enhances the overall scalability of the framework. Another key contribution of our work is the introduction of the **Weighted Clique Edge Cover (WCEC)** optimization problem. Empirical results demonstrate that solving WCEC enables effective reconstruction of hypergraphs from weighted graph representations, bridging the gap between pairwise data and higher-order relationships.

We hope our work inspires further exploration and advancements in this area of research. Once again, thank you for all the valuable input and thoughtful feedback, which have been instrumental in shaping this paper and moving this field forward.

---

### Meta-Review · Area_Chair_xpCs · 2024-12-21

**Metareview:**

The authors propose a learning-based generative model for large-scale hypergraphs (e.g., email networks), which have commonly been modeled using rule-based or mathematical models. Their approach reformulates the problem at the graph level, focusing on generating a realistic projected graph based on the ground-truth projected graph (also referred to as clique expansion). Subsequently, a realistic hypergraph is reconstructed from the projected graph. The experiments demonstrate that the proposed model yields hypergraphs with reasonably realistic structural properties.

The reviewers recognized the use of a machine-learning approach to large-scale hypergraph generation as an interesting and valuable direction, and they found the overall method design compelling.

However, they also raised several limitations, including limited scalability, marginal performance improvements over rule-based models, and areas where the presentation could be enhanced.

While no reviewer expressed strong enthusiasm for the submission, all reviewers leaned slightly toward acceptance. The meta-reviewer also believes that, as the first machine-learning-based approach to tackle an important problem, the merits of accepting the paper outweigh the drawbacks of rejecting it. Therefore, the meta-reviewer recommends acceptance, provided the authors diligently improve the paper based on the discussions.

**Additional Comments On Reviewer Discussion:**

The authors addressed most, though not all, of the concerns through extensive rebuttals.

---

### Decision · Program_Chairs · 2025-01-22

Accept (Poster)